# Co-Occurrence of Gram-Negative Rods in Patients with Hematologic Malignancy and Sinopulmonary Mucormycosis

**DOI:** 10.3390/jof10010041

**Published:** 2024-01-04

**Authors:** Stephanie L. Egge, Sebastian Wurster, Sung-Yeon Cho, Ying Jiang, Dierdre B. Axell-House, William R. Miller, Dimitrios P. Kontoyiannis

**Affiliations:** 1Center for Infectious Diseases, Houston Methodist Research Institute, Houston, TX 77030, USA; 2Division of Infectious Diseases, Department of Internal Medicine, Houston Methodist Hospital, Houston, TX 77030, USA; 3Department of Infectious Diseases, Infection Control and Employee Health, The University of Texas M.D. Anderson Cancer Center, Houston, TX 77030, USA; 4Division of Infectious Diseases, Department of Internal Medicine, Vaccine Bio Research Institute, College of Medicine, The Catholic University of Korea, Seoul 06591, Republic of Korea; 5Catholic Hematology Hospital, Seoul St. Mary’s Hospital, Seoul 06591, Republic of Korea

**Keywords:** mucormycosis, pulmonary infection, bacterial pneumonia, Gram-negative rods, co-infection, *Pseudomonas aeruginosa*, hematological malignancy

## Abstract

Both Mucorales and Gram-negative rods (GNRs) commonly infect patients with hematological malignancies (HM); however, their co-occurrence is understudied. Therefore, we retrospectively reviewed the records of 63 patients with HM and proven or probable sinopulmonary mucormycosis at MD Anderson Cancer Center (Houston, Texas) from 2000–2020. Seventeen out of sixty-three reviewed patients (27.0%) had sinopulmonary co-occurrence of GNRs (most commonly *Pseudomonas aeruginosa* and *Stenotrophomonas maltophilia*) within 30 days of a positive Mucorales culture or histology demonstrating Mucorales species. Eight of seventeen co-isolations of Mucorales and GNRs were found in same-day samples. All 15 patients with GNR co-occurrence and reported antimicrobial data had received anti-Pseudomonal agents within 14 days prior to diagnosis of mucormycosis and 5/15 (33.3%) had received anti-Stenotrophomonal agents. Demographic and clinical characteristics of patients with and without GNR co-occurrence were comparable. Forty-two-day all-cause mortality was high (34.9%) and comparable in patients with (41.2%) and without (32.6%) GNR detection (*p* = 0.53). In summary, over a quarter of heavily immunosuppressed patients with sinopulmonary mucormycosis harbored GNRs in their respiratory tract. Although no impact on survival outcomes was seen in a background of high mortality in our relatively underpowered study, pathogenesis studies are needed to understand the mutualistic interplay of GNR and Mucorales and their influence on host responses.

## 1. Introduction

Mucormycosis is a severe and frequently lethal opportunistic infection in susceptible hosts such as diabetics in ketoacidosis, patients undergoing hematopoietic cell or solid organ transplant, patients receiving corticosteroids, and patients with hematological malignancies (HM) [1]. The most common manifestation of mucormycosis in patients with HM is sinopulmonary infection [1,2]. Hallmarks of the pathogenesis of sinopulmonary mucormycosis are its rapid progression by invasion of epithelial barriers, propensity to cause tissue necrosis, and hematogenous dissemination [3]. In combination with the limited diagnostic modalities for early detection of mucormycosis and the insufficient therapeutic armamentarium, these pathogenetic hallmarks contribute to high mortality, especially in patients with sustained breaches of host defense (e.g., those with persistent neutropenia) [1,3,4].

Moreover, a growing body of experimental studies suggested the role of complex inter-microbial interactions between opportunistic molds and bacterial co-pathogens in the pathogenesis of respiratory infections. Specifically, opportunistic molds and Gram-negative rods (GNRs) can share the same niche in infected tissues of the sinopulmonary tract and cause pneumonia in immunocompromised hosts with HM [5]. Prior in vitro and some in vivo laboratory studies focusing on the interplay between GNRs and *Aspergillus* species revealed pleiotropic (synergistic, neutral, or antagonistic) interactions that mediate interspecies competition between molds and GNR [6]. These interactions can result in altered fungal growth, viability, biofilm formation, toxin production, and invasive capacity [6,7]. Mechanistic determinants of inter-kingdom interplay include competition for nutrients and iron (e.g., through the production of siderophores) [8,9], secreted proteins [10], and the release of volatile compounds [11]. Although more scarcely studied, competition between GNRs and Mucorales has been described and is partially mediated by iron sequestration due to the secretion of bacterial siderophores [12].

Despite these experimental insights, clinical data regarding the prevalence and prognostic significance of respiratory co-infections with both molds and GNRs in immunocompromised patients are limited, especially in the context of mucormycosis. Therefore, we herein studied the prevalence of the co-occurrence of Mucorales and GNR pathogens in patients with known HM and compared the clinical characteristics and outcomes of those patients with and without GNR co-occurrence.

## 2. Materials and Methods

### 2.1. Chart Review

We retrospectively reviewed 63 consecutive patients with HM and proven or probable sinopulmonary mucormycosis (EORTC/MSG criteria) [13], i.e., those with sinusitis and/or pneumonia, at MD Anderson Cancer Center (Houston, TX, USA) from 2000–2020 [14]. We reviewed demographic data, underlying malignancy, history of hematopoietic stem cell transplant, presence of neutropenia at mucormycosis diagnosis, site of mucormycosis and causative genus, bacterial cultures, use of anti-Pseudomonal and anti-Stenotrophomonal agents within 14 days of mucormycosis diagnosis, use of antifungal agents at the time of mucormycosis diagnosis, antifungal and surgical therapy of mucormycosis, intubation after mucormycosis diagnosis, and 42-day survival after mucormycosis diagnosis. Co-occurrence of GNRs was defined as a GNR-positive culture taken from either sinus or lung at the time of mucormycosis diagnosis or within 30 days after a positive Mucorales culture or histology demonstrating Mucorales species.

### 2.2. Statistical Analyses

Categorical variables were compared with Chi-squared or Fisher’s exact test, while continuous variables were compared using the Wilcoxson rank-sum test. Survival curves were compiled using the Kaplan–Meier method and differences in survival probabilities were determined using the Mantel–Cox test. All tests were 2-sided with a significance level of 0.05. Statistical analyses were performed using SAS version 9.4 (SAS Institute Inc., Cary, NC, USA) and Prism version 9 (GraphPad Software, Boston, MA, USA).

## 3. Results

The majority of Mucorales infections in both groups were seen in patients with underlying leukemia (56/63, 88.9%, Table 1). A small proportion of sinopulmonary Mucorales infections were found in patients with non-leukemia HMs, i.e., lymphoma (n = 5), multiple myeloma (n = 1), or myelodysplastic syndrome (n = 1). Most patients were neutropenic (absolute neutrophil count <500/mm^3^) at the time of mucormycosis diagnosis (40/63, 63.4%).

All patients had received antifungal agents at the time of mucormycosis diagnosis. Sixty out of sixty-three had received a single agent, most commonly voriconazole (35/63, 55.6%), followed by echinocandins (11/63, 17.5%; nine caspofungin, one anidulafungin, one micafungin), isavuconazole (8/63, 12.7%), posaconazole (4/63, 6.3%), and itraconazole (2/63, 3.2%). Three patients (4.8%) had received voriconazole plus caspofungin.

Seventeen of sixty-three patients (27.0%) had evidence of GNR co-occurrence. All 17 patients with GNR co-occurrence had leukemia, mostly acute myelogenous leukemia (8/17, 47.1%, Table 1). There were no statistical differences in GNR co-occurrence by malignancy type (Table 1). Similarly, the proportions of patients with a history of hematopoietic stem cell transplant were comparable between the two groups. At the time of the initial mucormycosis diagnosis, 52.9% and 67.4% of patients with and without GNR co-occurrence had neutropenia, respectively.

The most commonly isolated Mucorales genera were *Rhizopus* (n = 39), *Mucor* (n = 10), and *Rhizomucor* (n = 8). Additional genera included *Cunninghamella* (n = 3) and *Absidia* (n = 2). Distribution and prevalence of Mucorales genera were comparable among patients with and without GNR co-occurrence, except for *Absidia* infection, which had isolated occurrence in two patients without GNR co-occurrence (Table 1). Similarly, patients with and without GNR co-occurrence had largely comparable distributions of mucormycosis sites, except for a higher proportion of isolated rhinosinusitis cases in the GNR co-occurrence group compared to patients with mucormycosis alone (52.9% vs. 34.8%, Table 1). However, this trend did not reach statistical significance (*p* = 0.16).

GNR isolates included *Pseudomonas aeruginosa* (n = 8), *Stenotrophomonas maltophilia* (n = 6), *Achromobacter* species (n = 1), and Enterobacterales group species (n = 9) (Table 2). Five of seventeen patients (29.4%) had more than one GNR species recovered. GNR organisms were identified from sinus tissue or sinus aspirates (8/17, 47.1%), bronchoalveolar lavage, bronchial washing fluid or tracheal aspirate (5/17, 29.4%), and sputum (4/17, 23.5%, Table 2). In all but one case of Mucorales and GNR co-occurrence (16/17, 94.1%), the sites of bacterial isolation concurred with the site of mucormycosis (Table 2). Twelve of seventeen patients (70.6%) with GNR co-occurrence had sinopulmonary GNR growth within a week of mucormycosis diagnosis, including eight (47.1%) with co-occurrence in same-day samples (Table 2). Five of seventeen patients (29.4%) with GNR co-occurrence had multidrug-resistant organisms on culture (Table 2).

Nearly all patients (98.3%), including all patients with GNR co-occurrence and reported antimicrobial data, received one or more anti-Pseudomonal agents within 14 days of mucormycosis diagnosis (Table 2). Over half of all patients (52.5%) received therapy with an agent known to have activity against *S. maltophilia*. A higher proportion of patients without GNR co-occurrence (26/44, 59.1%) versus those with co-occurrence (5/15, 33.3%) had received anti-Stenotrophomonal antibiotics in the 14 days prior to mucormycosis diagnosis, but this trend was not statistically significant (*p* = 0.13, Table 1).

All patients received appropriate antifungal therapy for mucormycosis, mostly liposomal amphotericin B-based combination therapy (Table 1). The proportions of patients receiving monotherapy versus antifungal combination therapy were similar between the GNR co-occurrence group and patients with mucormycosis alone. Likewise, the frequency of surgical therapy for mucormycosis was comparable between the two groups (Table 1).

Unsurprisingly, mucormycosis had poor outcomes in our cohort of HM patients, regardless of GNR co-infection, with 42-day mortality rates of 41.2% (7/17) versus 32.6% (15/46) in patients with and without sinopulmonary GNR co-occurrence, respectively (*p* = 0.53, Table 1). Comparable survival outcomes in patients with and without GNR co-occurrence were confirmed by survival curve analysis (*p* = 0.42, Figure 1).

## 4. Discussion

In this retrospective analysis, we found that over 25% of patients (17/63) with sinopulmonary mucormycosis had evidence of GNR co-occurrence within the respiratory tract. Importantly, in nearly all cases, the site of GNR isolation was the same as the site of mucormycosis. The majority of GNR isolates were either *P. aeruginosa* or *S. maltophilia* species. Of the 59 patients with reported antimicrobial exposure data, 58 patients received anti-Pseudomonal therapy within 14 days prior to GNR isolation. Notably, all patients with *P*. *aeruginosa* co-occurrence had received at least one dose of anti-Pseudomonal therapy in the 14 days prior to GNR isolation; only two of these isolates demonstrated multi-drug resistance. Thus, *P*. *aeruginosa* co-occurrence does not seem to be associated with the emergence of resistant organisms. A greater number of patients without GNR co-occurrence received an antibiotic with activity against *S. maltophilia*, although this trend did not reach statistical significance. Since *S. maltophilia* was the second most common organism isolated from patients with GNR co-occurrence, the use of antimicrobials with activity against these bacteria may have, in part, influenced the culture results.

Importantly, *P. aeruginosa* and *S*. *maltophilia* are common colonizers of the respiratory tract as well as opportunistic pathogens [15,16]. In the immunocompromised host, these species are often pathogenic in patients with neutropenic fever and/or respiratory symptoms and imaging suggestive of lung infection (e.g., nodules, bronchial wall thickening, ground glass opacity, or consolidation on imaging) [17,18]. However, these findings may overlap with those of mucormycosis and thus may confound the clinical impression. Thus, defining the pathogenic role of GNRs in this setting is difficult, and further studies are needed to better understand the clinical significance of GNR and Mucorales co-occurrence from respiratory cultures.

As all patients demonstrated an immunocompromised state and there were high rates of neutropenia and respiratory failure, it is unclear whether additional host factors contributed to the co-detection of GNRs in the setting of sinopulmonary mucormycosis. It is possible that common underlying factors favoring pathogen virulence, such as increased tissue iron availability and dysfunctional host immune response, might have accounted for the co-occurrence of these opportunistic organisms [12,19].

Interestingly, we found no significant difference in 42-day mortality rates between patients with GNR co-occurrence and those with mucormycosis alone. Given that outcomes of mucormycosis are predominantly host-driven [14], it is likely that the immuno-pathogenetic impact of respiratory GNR co-pathogens was limited in our cohort of HM patients who had a significant burden of immunosuppression. Specifically, we hypothesize that indirect, host-mediated inter-kingdom synergies, as described for various co-infection scenarios involving molds and/or GNRs [20,21,22], might have limited immuno-pathogenetic significance in patients with severe underlying quantitative (i.e., profound cytopenias) or qualitative (e.g., significant blastemia) immune deficits.

This study has limitations. Given the retrospective nature, long study period, and many competing causes of death in these severely ill patients, we could not fully assess infection-attributable mortality and had to resort to all-cause mortality for outcome analysis. Due to changes in the medical record system over the 20-year review period and referrals of patients pretreated at other hospitals, data regarding the exact duration of prior anti-infective treatment was not available for some patients. Thus, correlations between the duration and timing of antibiotic therapy and co-occurrence of GNRs could not be assessed. Similarly, the intention of anti-infective drug use (i.e., prophylaxis, preemptive, or empirical) was difficult to elucidate solely based on retrospective chart review. Lastly, given the monocentric nature of this study, our results may not be generalizable to other institutions and host groups at risk for mucormycosis and inter-kingdom infections (e.g., patients with underlying COVID-19 infection).

In summary, this retrospective analysis in a sizeable cohort of HM patients with sino-pulmonary mucormycosis revealed that co-occurrence of GNRs and Mucorales was seen in over a quarter of patients but did not impact mortality outcomes of mucormycosis in our cohort. As the site of GNR isolation and mucormycosis was the same in nearly all cases, further mechanistic studies looking at interspecies competition in different host contexts (e.g., diabetic ketoacidosis versus active hematologic cancer) and clinical studies assessing response to therapy are needed to determine how co-occurring organisms might impact each organism’s virulence and overall infection outcome.

## Figures and Tables

**Figure 1 jof-10-00041-f001:**
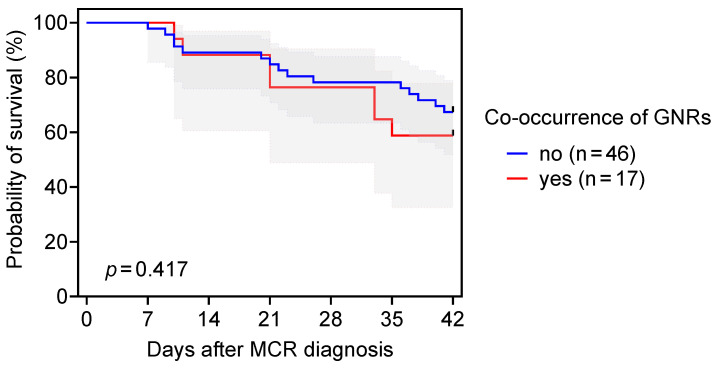
Survival curves for patients with hematologic malignancy and sinopulmonary mucormycosis (MCR), with and without co-occurrence of Gram-negative rods (GNRs). Error bands indicate 95% confidence intervals. Mantel–Cox log-rank test.

**Table 1 jof-10-00041-t001:** Demographics and clinical variables in patients with and without co-occurrence of Mucorales and Gram-negative rods.

Characteristics	All Patients (n = 63)	Co-Occurrence of Gram-Negative Rods and Mucorales (n = 17)	Without Co-Occurrence of Gram-Negative Rods (n = 46)	*p*-Value
Age (years), median (range)	52 (18–75)	52 (34–69)	50 (18–75)	0.21
Sex, male, n (%)	43 (68.3)	12 (70.6)	31 (67.4)	0.81
Neutropenic at time of Mucorales diagnosis, n (%)	40 (63.4)	9 (52.9)	31 (67.4)	0.29
History of stem cell transplant, n (%)	40 (63.4)	11 (64.7)	29 (63.0)	0.90
Underlying malignancy, n (%)				
Acute myelogenous leukemia	31 (49.2)	8 (47.1)	23 (50.0)	0.84
Acute lymphoblastic leukemia	14 (22.2)	5 (29.4)	9 (19.6)	0.50
Chronic lymphocytic leukemia	7 (11.1)	4 (23.5)	3 (6.5)	0.08
Lymphoma	5 (7.9)	0	5 (10.9)	0.31
Chronic myelogenous leukemia	4 (6.3)	0	4 (8.9)	0.57
Myelodysplastic syndrome	1 (1.6)	0	1 (2.2)	>0.99
Multiple myeloma	1 (1.6)	0	1 (2.2)	>0.99
Mucorales species isolated, n (%)				
*Rhizopus* spp.	39 (61.9)	12 (70.6)	27 (58.7)	0.39
*Mucor* spp.	10 (15.9)	3 (17.7)	7 (15.2)	>0.99
*Rhizomucor* spp.	8 (12.7)	1 (5.9)	7 (15.2)	0.43
*Cunninghamella* spp.	3 (4.8)	1 (5.9)	2 (4.4)	>0.99
*Absidia* spp.	2 (3.2)	0 (0.0)	2 (4.4)	>0.99
Unknown/Unclassified	1 (1.6)	0 (0.0)	1 (2.17)	>0.99
Site of mucormycosis, n (%)				
Rhinosinusitis	25 (39.7)	9 (52.9)	16 (34.8)	0.16
Pneumonia	24 (38.1)	5 (29.4)	19 (41.3)	0.56
Rhinosinusitis + pneumonia	9 (14.3)	3 (17.6)	6 (13.0)	0.69
Rhinosinusitis + cerebral involvement	3 (4.8)	0 (0.0)	3 (6.5)	0.56
Rhinosinusitis + orbital involvement	2 (3.2)	0 (0.0)	2 (4.4)	>0.99
Antibiotic(s) with activity against *P. aeruginosa* within 14 days of mucormycosis diagnosis ^a^, n (%)	58/59 * (98.3)	15/15 * (100.0)	43/44 * (97.7)	>0.99
Antibiotic(s) with activity against *S. maltophilia* within 14 days of mucormycosis diagnosis ^b^, n (%)	31/59 (52.5)	5/15 * (33.3)	26/44 * (59.1)	0.13
Intubation after mucormycosis diagnosis, n (%)	19 (30.2)	2 (11.8)	17 (37.0)	0.07
Antifungal therapy of mucormycosis				0.25
Liposomal amphotericin B monotherapy	9 (14.3)	8 (17.4)	1 (11.1)	
Combination therapy	54 (85.7)	16 (94.1)	38 (82.6)	
Surgical therapy of mucormycosis	32 (50.8)	10 (58.8)	22 (47.8)	0.57
Expired within 42 days, n (%)	22 (34.9)	7 (41.2)	15 (32.6)	0.53

^a^ Ciprofloxacin, levofloxacin, amikacin, meropenem, imipenem, piperacillin-tazobactam, cefepime, ceftazidime; ^b^ Levofloxacin, minocycline, tigecycline, ceftazidime, trimethoprim-sulfamethoxazole; * Numerators representative of patient data with a total of four exclusions (two from each group) due to unreported antimicrobial data.

**Table 2 jof-10-00041-t002:** Summary of clinical characteristics in leukemia patients with sinopulmonary co-occurrence of Gram-negative rods and Mucorales.

Age (y)	Sex	HM	Allo-SCT History	GNR TTD (d)	GNR(s)	MCR Genus	MCR Site	GNR Site	ETT, Trach	ANC <500	Death ≤42 d	Anti-PA Agent ^b^	Anti-SM Agent ^b^
39	M	AML	MUD	0	*S. maltophilia*	*Rhizopus*	RS	BAL		X		X	X
39	F	B-ALL	MUD	6	*S. maltophilia*, *E. coli*	*Rhizopus*	RS	ST			X	X	
68	M	AML	None	0	*S. maltophilia*	*Mucor*	P	TA				X	
64	M	CLL	MUD	#	*S. maltophilia*	*Rhizopus*	RS	SA		X	X	N.R.	N.R.
52	M	CLL	MUD	20	*S. maltophilia*	*Rhizopus*	P	SP				X	X
58	M	CLL	None	0	*P. aeruginosa*, *E. coli*, *K. pneumoniae*	*Rhizomucor*	RS + P	BW		X	X	X	X
46	F	CLL	MRD	0	*P. aeruginosa*, *E. cloacae* ^a^	*Cunninghamella*	RS + P	SP				X	
67	M	AML	MRD	0	*P. aeruginosa*, *S. maltophilia*	*Rhizopus*	RS + P	SP				X	
69	F	AML	None	6	*P. aeruginosa*	*Rhizopus*	RS	SA			X	X	
66	F	AML	MRD	0	*P. aeruginosa* ^a^	*Rhizopus*	RS	ST				X	
61	F	AML	None	13	*P. aeruginosa* ^a^	*Mucor*	P	BAL		X		X	
54	M	AML	None	3	*P. aeruginosa*	*Mucor*	P	SP		X	X	X	
43	M	AML	MUD	0	*P. aeruginosa*	*Rhizopus*	P	TA				N.R.	N.R.
34	M	B-ALL	MUD	28	*E. coli, E. cloacae*, *K. pneumoniae* ^a^	*Rhizopus*	RS	SA		X		X	
52	M	B-ALL	None	1	*E. coli*	*Rhizopus*	RS	SA		X	X	X	X
47	M	B-ALL	MUD	15	*E. coli*	*Rhizopus*	RS	SA	X	X	X	X	
45	M	B-ALL	MUD	0	*Achromobacter* spp. ^a^	*Rhizopus*	RS	ST	X	X		X	X

^a^ Denotes cultures with evidence of at least 1 MDR GNR. ^b^ Anti-Stenotrophomonal or anti-Pseudomonal agents were counted if at least one dose of known anti-Pseudomonal or anti-Stenotrophomonal antibiotics was given within 14 days prior to positive MCR culture/histopathology. # Time between MCR diagnosis and GNR culture was within 30 days but was not definitively known, as this patient was transferred from an outside hospital. **Abbreviations in headers:** d = days, HM = hematological malignancy, allo-SCT: allogenic stem cell transplant, MCR: Mucorales, GNR: Gram-negative rod(s), TTD: time to diagnosis, MDR: multi-drug resistant organism on Gram-negative culture, ETT, trach: endotracheal tube or tracheostomy placed in time frame between MCR and GNR culture positivity, PA: *P. aeruginosa*, SM: *S. maltophilia*, y = years. **Malignancy abbreviations:** AML: acute myelogenous leukemia, B-ALL: B-cell acute lymphocytic leukemia, T-ALL: T-cell acute lymphocytic leukemia, CLL: chronic lymphocytic leukemia, CML: chronic myelogenous leukemia. **Allo-SCT abbreviations:** MUD: matched unrelated donor, MRD: matched related donor. **MCR infection classification abbreviations:** RS: rhinosinusitis, P: pneumonia. **GNR site abbreviations:** BAL = bronchoalveolar lavage, BW = bronchial washing fluid, SA = sinus aspirate, ST = sinus tissue, SP = sputum, TA = tracheal aspirate.

## Data Availability

Data are contained within the article.

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
