# Peer review of "Co-Occurrence of Gram-Negative Rods in Patients with Hematologic Malignancy and Sinopulmonary Mucormycosis"

_jof, 2024, doi:10.3390/jof10010041_

Round 1

Reviewer 1 Report

Comments and Suggestions for Authors

Manuscript by Egge et al described the co-occurence of GNR in patients with sinopulmonary mucormycosis. The topic of the manuscript is very interesting and novel but I have some general comments for the authors that would need further clarification

Introduction - It would be good to expand upon the intro section a bit. E.g. importance of mucormycosis in patients with haematological malignances, risk factors, comorbidities and then maybe introduce the concept of co-ocurrence and clearly exlplain what is known so far.

Methods - details on ethical approval if required is missing

Discussion is very nice but I would like to see more info in the wither context of co-occurence of these microorganisms and mucorales. I really think the data is very interesting but the paper needs to be expanded a bit.

Author Response

Reviewer 1

Introduction - It would be good to expand upon the intro section a bit. E.g. importance of mucormycosis in patients with hematological malignances, risk factors, comorbidities and then maybe introduce the concept of co-occurrence and clearly explain what is known so far.

We thank the reviewer for this suggestion. We have added an introductory paragraph about mucormycosis, its risk factors, and key pathogenetic hallmarks (lines 38-48). We have also expanded the paragraph introducing the significance and mechanisms of polymicrobial sinopulmonary infections (lines 49-62).

Methods - details on ethical approval if required is missing.

This information has been provided in the footnote section (lines 244-247), as required by the official Journal of Fungi template.

Discussion is very nice, but I would like to see more info in the wider context of co-occurrence of these microorganisms and Mucorales. I really think the data is very interesting, but the paper needs to be expanded a bit.

We have expanded the Discussion section as suggested. Specifically, to provide a wider context, we have added a paragraph that discusses the role of host-mediated inter-kingdom synergies and its possibly limited significance in the context of severe underlying host immune dysfunction (lines 204-212). We have also expanded our discussion of key limitations of this study, partially inspired by the points raised by reviewer 2 (lines 213-224).

Reviewer 2 Report

Comments and Suggestions for Authors

Patients with blood neoplasmas and treated with haematopoietic stem cell transplants represent a high-risk group for bacterial viral and fungal infections. Of the latter, mucormycosis has become increasingly important in recent years. These infections are much rarer than bacterial, hence the material comprising 63 consecutive patients from one hospital should be considered significant. The authors of this study focused on the co-occurrence of mucormycosis and infection caused by gram negative bacilli. This is an interesting aspect, but in my opinion a few points need further clarification.

11.  The most important issue is whether patients have received antifungal and antimicrobial prophylaxis ( if so, which and how long).

22. To complete the characteristics of the study groups, I would suggest including data on antifungal treatment and possible surgery in Table 1.

33. The term 'sinopulmonary mucormycosis' is misleading, clearly suggesting simultaneous infection of the sinuses and lungs, whereas most often the infection involved only the lungs or only the sinuses. This should have already been clarified at the beginning of the article.

43. Similarly, the term " sinopulmonary GNR co-occurrence" is very general. Since the location of mucormycosis is given in Table 1, it is also useful to give the location of the bacterial infection (or isolation); it would be interesting to know whether in individual patients the bacterial and fungal pathogens occurred together or involved different niches.

Author Response

Reviewer 2

 #1. The most important issue is whether patients have received antifungal and antimicrobial prophylaxis (if so, which and how long).

As detailed in Materials & Methods and Table 1 & 2, we had already reviewed anti-pseudomonal and/or anti-stenotrophomonal agents within 14 days of mucormycosis diagnosis in our initial submission.

All patients had received antifungal agents at the time of mucormycosis diagnosis. The detailed breakdown has been added in lines 102-106.

Although we agree with the reviewer that duration of anti-infective therapy might affect GNR co-occurrence in mucormycosis patients, data is incomplete due to the retrospective nature of the study, changes in completeness of our documentation in our medical record system over time, and referrals of patients pretreated at other hospitals. Therefore, we do not have access to granular data for many patients. However, we believe that this limitation does not affect or change the main results and conclusion of this pilot study. Furthermore, the purpose of anti-infective drug usage (i.e., prophylaxis, preemptive, or empirical) is difficult to elucidate solely based on retrospective chart review. Therefore, we use the broader term “antibiotic/antifungal drug use” throughout the manuscript. Our original submission had already acknowledged these important limitations and we have now re-phrased them even more clearly (lines 216-221).

#2. To complete the characteristics of the study groups, I would suggest including data on antifungal treatment and possible surgery in Table 1.

Thank you for this suggestion. We added these variables to the Result section (lines 141-146) and Table 1. The percentage of patients who underwent surgery for the treatment of mucormycosis was comparable between the GNR co-occurrence group and patients with mucormycosis alone (58.8% vs. 47.8%, p=0.57). Similarly, proportions of patients receiving liposomal amphotericin B monotherapy versus antifungal combination therapy were similar between the GNR co-occurrence group and patients with mucormycosis alone (p=0.25).

#3. The term “sinopulmonary mucormycosis” is misleading, clearly suggesting simultaneous infection of the sinuses and lungs, whereas most often the infection involved only the lungs or only the sinuses. This should have already been clarified at the beginning of the article.   

Although the term “sinopulmonary mucormycosis” has been commonly used in the mycology literature by us and others to describe the continuum of partially overlapping mucormycosis manifestations in the respiratory tract, we agree with the reviewer that a definition is warranted to avoid the impression that all enrolled patients had simultaneous infection of both sites. We have added a clarification in lines 72-73. We have also added more granular data regarding the sites of mucormycosis in Table 1.

#4. Similarly, the term "sinopulmonary GNR co-occurrence" is very general. Since the location of mucormycosis is given in Table 1, it is also useful to give the location of the bacterial infection (or isolation); it would be interesting to know whether in individual patients the bacterial and fungal pathogens occurred together or involved different niches.

Thank you for raising this important point. We have reviewed the sites of GNR isolation and added this information to the Results section (lines 125-129) and Table 2. Importantly, sites of bacterial isolations concurred with the site of mucormycosis in 16 out of the 17 patients with co-occurrence of Mucorales and GN